# New 2– and 3–loop heavy flavor corrections to unpolarized and polarized deep-inelastic scattering[1]

**Jakob Ablinger[1], Johannes Blümlein[2⋆], Abilio De Freitas[2], Marco Saragnese[2], Carsten Schneider[1] and Kay Schönwald[3]**

**1** Johannes Kepler University Linz, Research Institute for Symbolic Computation (RISC),
Altenberger Straße 69, A-4040, Linz, Austria
**2** Deutsches Elektronen-Synchrotron DESY,
Platanenallee 6, D-15738 Zeuthen, Germany
**3** Institut für Theoretische Teilchenphysik Campus Süd,
Karlsruher Institut für Technologie (KIT), D-76128 Karlsruhe, Germany

⋆ Johannes.Bluemlein@desy.de

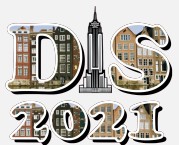
*Proceedings for the XXVIII International Workshop
on Deep-Inelastic Scattering and Related Subjects,
Stony Brook University, New York, USA, 12-16 April 2021*

## Abstract

**A survey is given on the new 2– and 3–loop results for the heavy flavor contributions to deep–inelastic scattering in the unpolarized and the polarized case. We also discuss related new mathematical aspects applied in these calculations.**

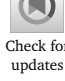
## 1 Introduction

The scaling violations of deep–inelastic structure functions provide a precise way to measure the strong coupling constant $\alpha_s(M_Z)$ [1–3]. This requires to calculate their $Q^2$ dependence both due to the massless and massive contributions at highest precision. Moreover, the structure functions also allow the precise measurement of the charm quark mass, $m_c$, [4]. At future facilities, like the EIC [5] or the LHeC [6, 7], operating at high luminosity, unpolarized and polarized structure functions can be measured at high precision to supplement and extend the present deep–inelastic world data. The massless QCD corrections are available to 3–loop order [8–16]. In the case of massive corrections, analytic results at the 3–loop level can currently only be obtained in the asymptotic approximation, $Q^2 \gg m_q^2$, with $Q^2$ the virtuality of the process and $m_q$ the heavy quark mass, cf. [17], which, however, are accurate to $\sim 1\%$ for the structure function $F_2(x, Q^2)$ in the heavy quark region of smaller values of $x$ for $Q^2/m_q^2 \gtrsim 10$

---

[1]Also contribution to the Proceedings of RADCOR 2021, Tallahassee, FL, May 2021.

already. This region should be chosen also to avoid higher twist effects, requiring at least the cuts $W^2 > 15$ GeV$^2$, $Q^2 > 10$ GeV$^2$, [18].

After having obtained a series of Mellin moments for massive 3–loop operator matrix elements (OMEs) in 2009 [19,20] the systematic calculation of the different OMEs contributing in the unpolarized and polarized case for the massive Wilson coefficients and the OMEs contributing to the matching conditions in the variable flavor number scheme (VFNS) has been started for the single mass [10,21–34] and the 2–mass contributions [35–38].

At 2–loop order the unpolarized and polarized non–singlet and pure singlet contributions have been calculated for the full kinematic region and analytic results have been derived for the power corrections in Refs. [17,39–42] to the structure functions $F_2, F_L$ and $g_1$.

The proceedings has the following structure. In Section 2 we describe the status of the calculation of the massive 3–loop OMEs and discuss the calculation methods used in Section 3. Recent results in the single mass and 2–mass cases are reported in Sections 4 and 5. Section 6 contains the conclusions.

## 2 Status of the massive OME calculations

In the leading twist approximation deep–inelastic structure functions have the representation

$$F_{2,L}(x, Q^2) = \sum_{i,q} \mathbb{C}_{2,L}^{(i)}\left(x, \frac{Q^2}{\mu^2}, \frac{m_q^2}{\mu^2}\right) \otimes f_{(i)}(x, \mu^2), \tag{1}$$

with $\otimes$ the Mellin convolution. A corresponding relation holds for the polarized structure function $g_1(x, Q^2)$. Here $\mathbb{C}^{(i)}$ denotes the Wilson coefficient related to the parton density $f_{(i)}$, with $\mu$ the factorization scale and $m_q$ the heavy quark masses, $m_q = m_c, m_b$, $x$ denotes the Bjorken variable. The Wilson coefficients can be decomposed into the massless, $C$, and massive contributions, $H$,

$$\mathbb{C}_{2,L}^{(i)}\left(x, \frac{Q^2}{\mu^2}, \frac{m_q^2}{\mu^2}\right) = C_{2,L}^{(i)}\left(x, \frac{Q^2}{\mu^2}\right) + H_{2,L}^{(i)}\left(x, \frac{Q^2}{\mu^2}, \frac{m_q^2}{\mu^2}\right). \tag{2}$$

At large scales $Q^2$ the heavy flavor Wilson coefficients have the representation [17]

$$H_{2,L}^{(i)}\left(x, \frac{Q^2}{\mu^2}, \frac{m_q^2}{\mu^2}\right) = \sum_{j,q} C_{(j),2,L}\left(x, \frac{Q^2}{\mu^2}\right) \otimes A^{ij}\left(x, \frac{m_q^2}{\mu^2}\right). \tag{3}$$

The leading order (LO) heavy flavor corrections were calculated for neutral and charged current interactions in [43–52] and numerically at next-to-leading order (NLO) in [53–57]. Analytic results at NLO have been obtained in complete form for the non–singlet and pure singlet corrections in [17,39–41] and in the asymptotic case $Q^2 \gg m_q^2$ in Refs. [17,27,39,58–63].

At 3–loop order only results in the asymptotic case have been calculated at present. In the single mass case the unpolarized OMEs for all OMEs $\propto N_F$ and the complete results for $A_{qq,Q}^{\text{NS}}$, $A_{qg,Q}, A_{qq,Q}^{\text{PS}}, A_{Qq}^{\text{PS}}, A_{gq,Q}$ [2] have been calculated in Refs. [10,21,22,25,64] and those $\propto T_F^2$ for $A_{gg,Q}$ in [23]. All logarithmic corrections were computed in [26]. For the pure $N_F$ contributions and both in the non–singlet case and for $A_{gq,Q}$ the OMEs can be expressed by harmonic sums [65,66] or harmonic polylogarithms [67] only. In the pure singlet case also generalized harmonic sums [68,69] contribute. In $z$ space harmonic polylogarithms of argument $z$ do not

---

[2]In the non–singlet case we have also calculated the OMEs for transversity.

span the result, unless one allows also for the argument $1-2z$. For the $T_F^2$ terms of $A_{gg,Q}$ also nested finite binomial sums contribute, leading to root–valued iterated integrals [70]. Rather involved binomial structures also occur in the case of $A_{Qg}$ [71]. The OME $A_{Qg}$ also contains iterative non–iterative integrals [72], containing complete elliptic integrals. The first order factorizing contributions to $A_{Qg}$ have been calculated in [31]. Phenomenological predictions in the non–singlet case have been given for the the structure functions $xF_3$ and $F_L^{W^+-W^-}(x,Q^2)$ and $F_2^{W^+-W^-}(x,Q^2)$ in [28, 30].

In the polarized case single mass contributions have been computed for the OMEs $A_{qq,Q}^{NS}$, $A_{qg,Q}$, $A_{qq,Q}^{PS}$, $A_{Qq}^{PS}$ and $A_{gq,Q}$ in Refs. [22, 32–34] and for all logarithmic contributions in [33]. Here the same mathematical structures as in the unpolarized case contribute. Phenomenological predictions in the non–singlet case for $g_1(x,Q^2)$ were given in [29].

The 2–mass corrections contributing from 3–loop order onward were calculated in the unpolarized case for $A_{qq,Q}^{NS}, A_{qg,Q}$ in [35], $A_{gg,Q}$ [36] and $A_{Qq}^{PS}$ [35] and analogously in the polarized case [34, 37, 38, 38]. The analytic results in $z$ space can be expressed by iterative integrals over root–valued letters, also parameterized by the mass ratio $m_c^2/m_b^2$.

Because of the missing hierarchy between $m_c^2$ and $m_b^2$, one has to decouple both heavy quark effects together in a variable flavor number scheme [59], generalizing the single–mass variable flavor number scheme [35, 73].

Further calculations concern the missing terms for the OMEs $A_{gg,Q}$ in the single mass case and for $A_{Qg}$ in the single and 2–mass case. For $A_{Qg}$ there is a first phenomenological representation in [74] based on only five Mellin moments calculated by us in Ref. [19, 20].

# 3   Calculation of the 3–loop OMEs

A survey on the calculation methods for the 3–loop massive OMEs has been given in Ref. [75]. Massive OMEs contain, beyond the usual Feynman rules, those for the twist–2 local operators, cf. [19, 20]. To apply integration-by-parts (IBP) techniques [76, 77] one needs to resum these operators into propagators [78]. In the topological simple cases we perform the calculation of the 3–loop integrals directly, using hypergeometric techniques [79–86]. More complex integrals are calculated using the method of first order factorizing differential and difference equations [71, 87]. From the IBP reductions one obtains systems of differential equations which can be mapped into systems of difference equations, allowing to calculate a large number of Mellin moments for the master integrals and the OMEs by using the method of arbitrary large moments [88]. Having generated a sufficient number of moments the method of guessing [89–91] allows to find the difference equations for the respective color and zeta factors

of the different OMEs. The number of these moments in the case of $A_{Qg}^{(3)}$ is very large. Already for the $T_F^2$ terms one needs $\sim 7500$ moments [31].

If the difference equations obtained factorize to first order, the difference ring techniques [92–104] implemented in the package `Sigma` [105, 106] are sufficient to find the final solution in Mellin $N$ space. The $z$ space solutions can be obtained by the techniques implemented in the package `HarmonicSums` [65–67, 69, 70, 107–117] in terms of iterated integrals over certain alphabets, which are found algorithmically. All massive 3–loop OMEs except those in the single and 2–mass case of $A_{Qg}^{(3)}$ belong to this class and have been solved by now.

In the following we describe some recent results in calculating single and 2–mass contributions at the 2– and 3–loop level.

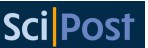

Figure 1: The ratios of the power expanded structure function to the complete structure function, $R_{2,q}^{(1)}$ (left) and $R_{L,q}^{(1)}$ (right), as a function of $\chi = 1/\kappa = Q^2/m_q^2$ for different values of $z$ gradually improved with $\kappa$ suppressed terms. Dotted lines: asymptotic result; dashed lines: $O(m_q^2/Q^2)$ improved; solid lines : $O((m_q^2/Q^2)^2)$ improved. From Ref. [41].

## 4 Single mass contributions

We consider first the successive inclusion of power corrections in $m_q^2/Q^2$ to the asymptotic result for the heavy flavor contributions to the structure functions $F_2(x, Q^2)$ and $F_L(a, Q^2)$ at NLO. The power corrections are of clear importance for $F_L$, since the pure asymptotic terms describe the heavy flavor corrections for $Q^2/m_q^2 \gtrsim 800$ only. Also in the case of $F_2$ for larger values of $z$ an improved description is obtained. The results for the pure–singlet contribution to the polarized structure function $g_1(x, Q^2)$ [42] are similar. The inclusion of the power corrections in the pure–singlet case in the unexpanded expressions leads to new iterative integrals over a corresponding alphabet, cf. [41, 42]. Expanding in the ratio $m_q^2/Q^2$ leads to harmonic polylogarithms again. The situation is simpler in the non–singlet case, where classical polylogarithms with more complicated arguments suffice in the general case, cf. e.g. [40]. Note that

the corresponding Wilson coefficients are not the ones given in [17], but need an extension.

The most important calculation for the future consist in the analytic computation of the OME $A_{Qg}^{(3)}$. The terms $\propto N_F$ were calculated in [21]. Based on 1000 Mellin moments all contributions, but the pure rational and $\zeta_3$ terms were calculated, since these can be obtained from difference equations which are factorizing at first order, which is not the case for the former terms, [31]. Based on 8000 moments we obtained the difference equations for all missing terms $\propto T_F^2$. In the $T_F$-case one will need more moments to find the corresponding difference equations. In the $T_F^2$ case the difference equations have the following characteristics for degree d and order o:

$$
\begin{aligned}
T_F^2 C_A &: \quad (d; o) = (1407; 46) \\
T_F^2 C_A \zeta_3 &: \quad (d; o) = (323; 24) \\
T_F^2 C_F &: \quad (d; o) = (654; 27) \\
T_F^2 C_F \zeta_3 &: \quad (d; o) = (283; 14).
\end{aligned}
$$

The first difference equation is more voluminous than the largest occurring in guessing the largest contribution to the 3–loop massless Wilson coefficient in Ref. [91]. For the 3–loop massive form factor, [118], one difference equation of $(d, o) = (1324; 55)$ has been obtained.

The separate analysis of the difference equations for the rational and $\zeta_3$ $T_F^2$ cases showed that the associated differential equations develop exponential singularities in the region $z \in [0, 1]$, although the complete solution is regular. Indeed, the differential equation for the purely rational term develops a $\zeta_3$ factor asymptotically such that both the singularities cancel. This requires to deal with both difference equations at once, despite the fact, that one can establish them only separately.

Starting from the difference equations in Mellin $N$ space one may find Laurent series solutions of the associated differential equations around $z_0 = 1$ up to a finite upper power in $z$. This expansion is possible also around any other regular point $z_0 \in [0, 1]$, [119]. All these expansions have a finite convergence radius and several expansions are necessary to map out the interval $z \in [0, 1]$ in terms of overlapping expansions. In this way one obtains an approximate analytic solution, which may be tuned to any accuracy. Given the fact that also all the other special functions need numerical representations, this approach has the advantage of already obtaining the numerical representation. The generation of a high number of moments for the $T_F$ terms is underway.

In Figure 2 we summarize the different contributions to the currently known charm quark QCD corrections up to 3–loop order to the structure function $F_2$ at the scale $Q^2 = 100$ GeV$^2$.

## 5   2–mass contributions

Irreducible 2–mass contributions emerge first at 3–loop order and lead to a change of the variable flavor number scheme [35]. Reducible contributions imply 2–mass contributions already at NLO [73]. In Figure 3 we illustrate the 2–mass effects on the singlet and bottom quark contribution. In the singlet case the effect amounts to 1% and in the case of the $b$-quark distribution of 4–5%.

We have also calculated the $O(\alpha_s^3)$ heavy flavor corrections normalized to the next-to-next-to-next-to leading order (N$^3$LO) flavor non–singlet structure functions $F_2^{\mathrm{NS}}(x, Q^2)$ and $g_1^{\mathrm{NS}}(x, Q^2)$ in the case of scheme–invariant evolution [122]. Here one considers the evolution

$$
F_2^{\mathrm{NS}}(x, Q^2) = E_{\mathrm{NS}}(x, Q^2, Q_0^2) \otimes F_2^{\mathrm{NS}}(x, Q_0^2) \tag{4}
$$

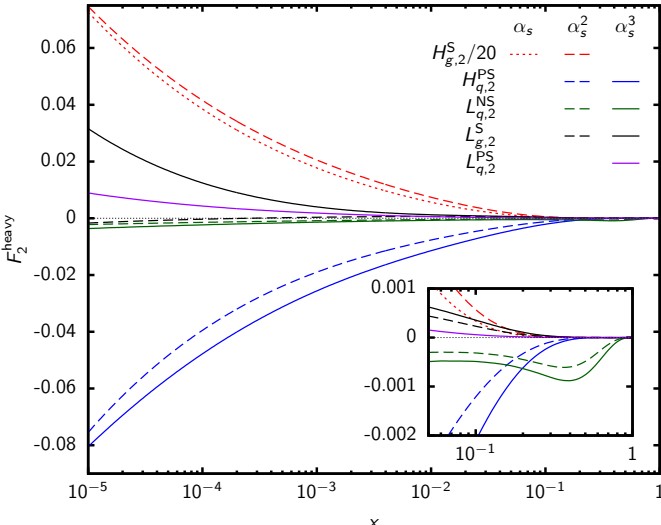

Figure 2: The different single mass heavy flavor contributions to the structure function $F_2$ at $Q^2 = 100\,\text{GeV}^2$. From [120].

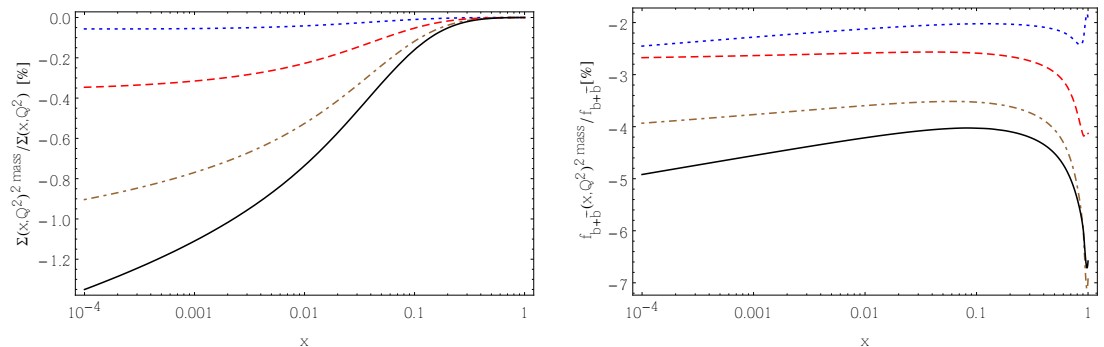

Figure 3: The ratio of the 2-mass contributions to the singlet parton distribution $\Sigma(x, Q^2)$ in the on-shell scheme. Dash-dotted line: $Q^2 = 30$ GeV$^2$; Dotted line: $Q^2 = 30$ GeV$^2$; Dashed line: $Q^2 = 100$ GeV$^2$; Dash-dotted line: $Q^2 = 1000$ GeV$^2$; Full line: $Q^2 = 10000$ GeV$^2$. For the PDFs the next-to-next-to leading order (NNLO) variant of ABMP16 with $N_f = 3$ flavors was used [121]. Form [73].

from a starting scale $Q_0^2$ to $Q^2$, where $E_{\text{NS}}(x, Q^2, Q_0^2)$ denotes a scheme–invariant evolution operator and the input distribution function $F_2^{\text{NS}}(x, Q_0^2)$ is measured experimentally. The evolution operator also depends on the masses $m_c$ and $m_b$. The results for the starting scale $Q_0^2 = 10$ GeV$^2$ are shown in Figure 4.

These corrections amount to $O(1\%)$ and are important in future factorization–scheme invariant measurements from the scaling violations of these structure functions at high luminosity facilities like the EIC or the LHeC.

## 6 Conclusions

Most of the massive 3–loop OMEs have been calculated in the single and 2–mass case. They contribute to the (2–mass) variable flavor number scheme and the heavy flavor Wilson coefficients in unpolarized and polarized deep–inelastic scattering in the asymptotic region $Q^2 \gg m_q^2$.

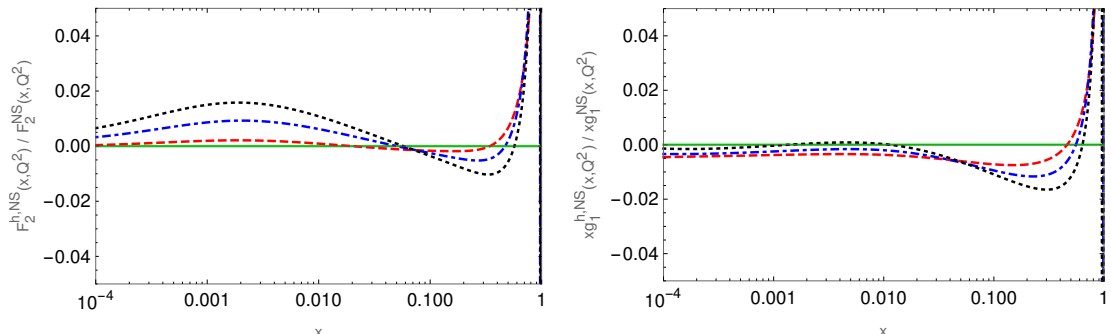

Figure 4: Left: The relative contribution of the heavy flavor contributions due to $c$ and $b$ quarks to the structure function $F_2^{\mathrm{NS}}$ at N³LO; dashed lines: 100 GeV²; dashed-dotted lines: 1000 GeV²; dotted lines: 10000 GeV². Right: The same for the structure function $x g_1^{\mathrm{NS}}$ at N³LO. From [122].

All quantities which can be described by first order factorizing difference equations have been computed. For the remaining OMEs the determination of their difference equations is underway. A method has recently been developed that can solve these equations as well. In all the computations extensive use has been made of the methods of arbitrary high Mellin moments, the method of guessing, and of difference ring theory to solve the respective physical problems. In course of these computations a series of different function spaces has been found to perform different intermediary steps of the calculation and to find a minimal analytic representation of the final results.

At 2–loop order analytic results have been derived for the non–singlet and pure–singlet Wilson coefficients in the whole kinematic region. The 2–mass corrections are quantitatively as important as the $O(T_F^2)$ contributions in the single mass case. The $O(T_F)$ contributions to the 3-loop anomalous dimensions have been calculated as by-product of the calculation of the massive OMEs and they agree with the results of previous calculations. The calculation methods have also being applied to higher order massive QED corrections for the initial state radiation to observables in $e^+ e^-$ annihilation [123].

## Acknowledgments

This project has received funding from the European Union's Horizon 2020 research and innovation programme under the Marie Skłodowska–Curie grant agreement No. 764850, SAGEX and from the Austrian Science Fund (FWF) grant SFB F50 (F5009-N15).

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
