# Peer review of "New 2- and 3-loop heavy flavor corrections to unpolarized and polarized deep-inelastic scattering"

_SciPost Physics Proceedings, doi:SciPost Phys. Proc. 8, 137 (2022)_

## Round 1 · Referee Report · Anonymous (Referee 1) · 2022-3-1

Report

Good results. The report requires improvements.

Suggested changes: 1. Adding further details in the abstract (from the conclusion). 2. Citation numbers are out of order. 3. Restructure the last few paragraphs of Sec. 1. 4. The report should be proofread by a native English speaker before publishing. 5. Fig 2, the contrast between dark green and black is not distinct. 6. Inconsistent use of two–mass and 2-mass. 7. Fig4, please define and explain N^3LO 8. What is the initial value for Q^2_0? 9. Fig 3, caption, the end of line #1: sigma (x, Q , need to be corrected. 10. Suggest including a diagram demonstrating: single mass and two mass interactions for both 2- and 3-loop contributions. 11. Fig 3 makes sense, but why is the y scale (ratio of the 2mass contribution and structure-functions) negative? Especially, the unit is in %.

  • validity: -
  • significance: -
  • originality: -
  • clarity: -
  • formatting: -
  • grammar: -

Author:  Kay Schoenwald  on 2022-03-03  [id 2262]

(in reply to Report 1 on 2022-03-01)
Category:
answer to question
reply to objection
correction

  1. We do not think the abstract needs to be extended, since it is fully clear.
  2. We do not find references which are out of order.
  3. We do not see a reason to restructure the last paragraphs.
  4. We disagree.
  5. The figure is reproduced from the original publication and there is copyright.
  6. Changed to 2--mass .
  7. Definition added in main text.
  8. Information added in main text.
  9. Fixed.
  10. The 2-mass corrections to the parton distributions are a 2-loop effect and can be negative. This is generally known for decades. The ratio is correctly given in percent.
  11. Due to the wealth of problems discussed a large number of diagrams would be needed. We think Feynman diagrams can be looked up in the cited references.

Anonymous on 2022-03-03  [id 2264]

(in reply to Kay Schoenwald on 2022-03-03 [id 2262])

With the newest changes, recommend for acceptance.

---

## Editorial Decision

published